# A Kernel Perspective on Few-Shot Adaptation of Large Vision-Language Models

## Abstract

The growing popularity of Contrastive Language-Image Pretraining (CLIP) has led to its widespread application in various visual downstream tasks. To enhance CLIP's effectiveness, efficient few-shot adaptation techniques have been widely adopted. Among these approaches, training-free methods, particularly caching methods exemplified by Tip-Adapter, have gained attention for their lightweight adaptation without the need for additional fine-tuning. In this paper, we revisit Tip-Adapter from a kernel perspective, showing that caching methods function as local adapters and are connected to a well-established kernel literature. Leveraging this insight, we offer a theoretical understanding of how these methods operate and suggest multiple avenues for enhancing over the Tip-Adapter baseline. Notably, our analysis shows the importance of incorporating global information in local adapters. Therefore, we subsequently propose a global method that learns a proximal regularizer in a reproducing kernel Hilbert space (RKHS) using CLIP as a base learner. Our method, that we call ProKeR (Proximal Kernel ridge Regression), has a closed form solution and achieves state-of-the-art performance across 11 datasets in the standard few-shot adaptation benchmark.

## 1 Introduction

Large scale vision-language models (VLMs) trained with contrastive learning have gained an increasing attraction in the recent years [41; 21]. These models have shown remarkable performance across a wide range of tasks such as classification [41], segmentation [30] and video understanding [29]. CLIP is one of the most established VLMs [41] offering remarkable zero-shot capabilities in various downstream tasks. It operates by using only the class label within a textual prompt such as "A photo of a {CLASS}" where {CLASS} is the groundtruth text label for each class. However, it is well-known that the zero-shot performance of CLIP is limited in scenarios with large domain shift from the pre-training distribution [10; 34]. To further improve CLIP's generalization, multiple follow-up works proposed to include few-shot data [66; 12; 62] which has shown remarkable performance gains compared to zero-shot CLIP.

Few-shot adaptation of CLIP can be categorized into two types of methods based on whether they require fine-tuning on the few-shot samples or not. Among fine-tuning based methods, prompt learning involves learning continuous tokens instead of hand-crafted templates in CLIP as proposed by CoOp [66] and CoCoOp [65]. Additionally, adapter-based fine-tuning methods operate in the features space to train their classifiers [12; 28]. Despite their promising performance on downstream tasks, fine-tuning methods require additional training costs to learn the new learnable parameters. Tip-Adapter proposed a training-free few-shot adaptation alternative [62]. Using a caching mechanism, Tip-Adapter captures knowledge from the few-shot samples without additional fine-tuning and ensembles it with zero-shot CLIP. This cache model has shown significant improvement over zero-shot performance which has led to follow-up works exemplified by APE [67] and Tip-X [53].

In order to understand caching effectiveness and limitations, we undertake a theoretical analysis to explore the nature of Tip-Adapter. We begin by demonstrating that the adaptation term of Tip-Adapter is a modified version of the well-known Nadaraya-Waston (NW) estimator [33], a local nonparametric regression method that allows Tip-Adapter to effectively capture various distributions. However, the NW estimator is also known to be strongly biased [25; 36]. To mitigate this bias, we leverage existing tools from the literature of kernel methods. An effective extension to NW estimator is locally

linear regression [46]. By fitting a local linear regression around each test point using a closed-form solution, we significantly improve the performance of Tip-Adapter. However, the parameters are estimated locally, which is prone to overfitting in high-dimensional problems [36; 16].

Our analysis shows that caching methods can be understood as local nonparametric regression methods regularised through CLIP pointwise zero-shot predictions. However, this regularization only acts locally and doesn't provide any global information about the few-shot task. Conversely, recent training-based methods rely on global regularizers to incorporate global information [50]. Hence, we ask ourselves the question, *how can we leverage global regularizers for few-shot adapters while conserving the benefits of training-free methods?* In order to design such a global regularizer, we devise two important design choices. Firstly, we restrain the hypothesis space of the learned function to be a reproducing kernel Hilbert space (RKHS). Secondly, using the RKHS norm, we introduce a proximal regularization term to ensure that the obtained solution is close to the base predictor i.e. $f_{\text{clip}}$. Thanks to the properties of the RKHS, minimizing the difference between two functions using the RKHS norm ensures that they are close pointwise. Our method ProKeR provides a more effective way to preserve prior knowledge from the zero-shot predictor and maintains the expressive capacity of the learned functions. Through extensive experiments, we show the effectiveness of our method ProKeR which achieves consistent gains over state-of-the-art training-free methods on standard few-shot classification benchmarks [62] with an absolute average improvement of $3.94\%$ accuracy. In addition, we highlight the robustness and generalizability of the proposed method across different architectures and on out-of-distribution datasets.

**Summary of contributions:**

1. We frame the cache model of Tip-Adapter as a Nadaraya-Waston estimator, a classical kernel regression method and provide a theoretical understanding of how Tip-Adapter operates.

2. Under this new perspective, we propose multiple improvements for Tip-Adapter either using a closed-form local linear regression fit or by incorporating global information.

3. We propose ProKeR, a training-free method that leverages global regularization in a RKHS. Through extensive experiments, ProKeR outperforms existing methods and sets a new state-of-the-art on standard few-shot classification benchmarks.

## 2 RELATED WORK

### 2.1 VISION-LANGUAGE PRE-TRAINED MODELS

In recent years, visual-language models (VLMs) have gained an increasing popularity. These methods, exemplified by CLIP [41], DeCLIP [27] and ALIGN [21], employ a contrastive learning framework to learn a vision encoder and a language encoder with a shared representation space between text and images. Trained on large-scale datats of image-text pairs, VLMs have shown remarkable zero-shot capabilities on downstream tasks without additional fine-tuning, paving the way for open vocabulary recognition [14]. VLMs have been extended to few-shot classification [66] as well as other tasks beyond image classification such as video understanding [29; 54], image segmentation [30; 52], image generation [43] and 3D reconstruction [63; 7].

### 2.2 FEW-SHOT ADAPTATION

Few-shot adaptation methods in the context of classification can be categorized into prompt learning and adapter based approaches. Inspired from the recent advances in natural language processing [26], prompt learning aims to learn effective global text or visual prompts for the downstream tasks [66; 65; 23; 2; 61; 58; 47; 6; 45; 49]. Although prompt learning methods have brought significant improvements over the zero-shot baseline, they require back-propagating through the entire text encoder and require having access to the text encoder [38]. On the other hand, adapter-based approaches operate in the feature space and do not require having access to the pre-trained model weights. We distinguish two families of adapter-based approaches. The first one is training-based methods [28; 12; 60] which either train a linear layer [28; 60] or a two-layer MLP such as CLIP-Adapter [12] to perform residual feature blending of the zero-shot classifier.

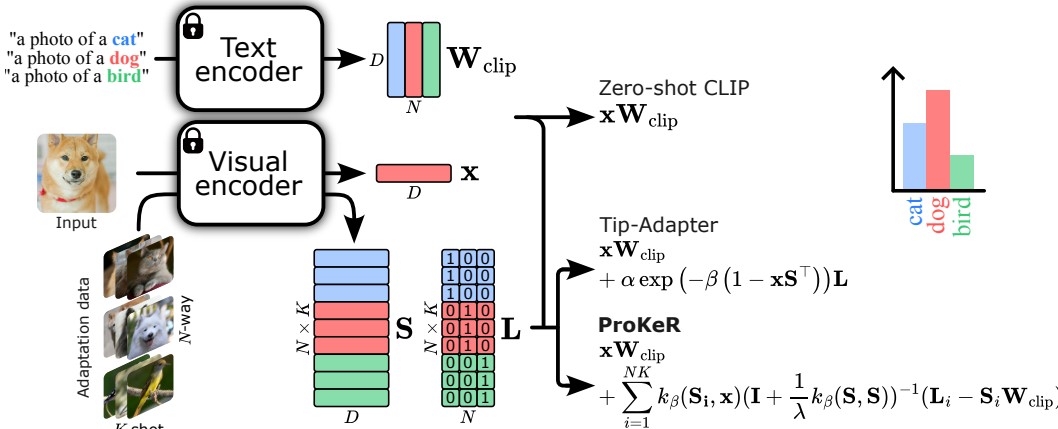

Figure 1: **Overview of our training-free method ProKeR**. While Tip-Adapter builds a key-value cache model using the few-shot samples, ProKeR incorporates a proximal global regularization based on the zero-shot predictor in a reproducing kernel Hilbert space (RKHS). This allows the use of a richer model without overfitting on the few-shot data.

While fine-tuning adapters have achieved a good level of performance, they still require additional training time which is impractical for limited resources and might suffer from the caveats of gradient optimization. To alleviate these issues, Tip-Adapter [62] emerges as a training-free solution based on a key-value cache model. Built on Tip-Adapter, multiple caching methods have been proposed. APE [67] includes a feature selection step to the cache model. CaFo [64] uses an ensemble of foundation models for the cache model and augments text prompts using a Large Language model. SuS-X and its module Tip-X [53] first generates the few-shot set using Stable Diffusion then uses the inter-modal similarities between the test images and the few-shot set. More recently, GDA proposes to use a Linear Discriminant Analysis to set a strong baseline for training-free methods [55]. Despite the good level of performance, there is currently no theoretical framework for better understanding the motivation behind caching models.

## 3 METHOD

We expose in this section the details of the proposed method. Our starting point consists in framing Tip-Adapter as a kernel method. Thanks to this new perspective, we develop multiple improvements for it. Conclusively, we propose ProKeR, a method that introduces a global proximal regularization in a reproducing kernel Hilbert space (RKHS).

### 3.1 TIP-ADAPTER AS A NADARAYA-WATSON ESTIMATOR

As illustrated in Fig. 1, let $\mathbf{x} \in \mathbb{R}^D$ be the features of an input query image extracted using the visual encoder of CLIP, $\mathbf{S} \in \mathbb{R}^{NK \times D}$ the visual features of the training set and $\mathbf{L} \in \mathbb{R}^{NK \times D}$ the associated matrix of one-hot labels where $N$ is the number of classes and $K$ is the number of shots per class. Let $\mathbf{W}_{\text{clip}} \in \mathbb{R}^{D \times N}$ be the text prototypes of the classes extracted with the text encoder using the standard hand-crafted templates in the form of "a photo of a {CLASS}" [62; 66; 28]. The zero-shot predictor from CLIP is defined as $f_{\text{clip}} : \mathbf{x} \mapsto \mathbf{x}\mathbf{W}_{\text{clip}}$.

To alleviate $f_{\text{clip}}$ prediction errors due to lack of generalization, Tip-Adapter [62] utilizes a cache model to learn knowledge from the few-shot samples. The predicted logits can be formulated as:

$$\phi_{\text{Tip}}(\mathbf{x}) = f_{\text{clip}}(\mathbf{x}) + \alpha \exp\left(-\beta\left(1 - \mathbf{x}\mathbf{S}^\top\right)\right)\mathbf{L} \tag{1}$$

where $\beta$ is a smoothing scalar and $\alpha$ controls the balance between textual features and few-shot images. Note that since the features in CLIP are normalized (i.e. $||\mathbf{x}||_2 = 1$), we can rewrite

equation 1 as:

$$\phi_{\text{Tip}}(\mathbf{x}) \;=\; f_{\text{clip}}(\mathbf{x}) + \alpha \sum_{i=1}^{NK} \exp\left(-\frac{\beta}{2}\|\mathbf{S}_i - \mathbf{x}\|_2^2\right)\mathbf{L}_i \tag{2}$$

where $\mathbf{S}_i$ is the $i$-th few-shot sample. Interestingly, the right term of Tip-Adapter can be seen as a modified version of the well-known Nadaraya-Watson (NW) estimator [33] with a Radial Basis Function (RBF) kernel [37]:

$$\phi(\mathbf{x}) \;=\; \frac{\sum_{i=1}^{NK} k_\beta(d(\mathbf{x}, \mathbf{S}_i))\mathbf{L}_i}{\sum_{i=1}^{NK} k_\beta(d(\mathbf{x}, \mathbf{S}_i))} \tag{3}$$

$$\text{where} \quad k_\beta(u) \;=\; \left(\frac{\beta}{2}\right)^D \exp(-\frac{\beta}{2}u) \tag{4}$$

where $d$ is the distance between the query image and the shots in the feature space. In essence, when $d$ is the Euclidean distance, as in equation 2, the adaptation term of Tip-Adapter is a nonparametric regression, obtained by performing a smooth and locally weighted average of the one-hot labels of the few-shot samples using a kernel function, which quantifies the similarity between the query image and the shots.

## 3.2 TRAINING-FREE FEW-SHOT ADAPTERS AS A BAYES OPTIMAL MAPPING

We formulate the few-shot visual-language adaptation as a Bayes optimal mapping [17] associated to the following pointwise risk.

$$R(\mathbf{x}, \phi(\mathbf{x})) = \mathbb{E}_{Y|X}[s(Y, \phi(X)) + \mathcal{R}_{\text{clip}} \mid X = \mathbf{x}] \tag{5}$$

where $s$ is a cost function and $\mathcal{R}_{\text{clip}}$ is a regularization term using CLIP prediction independent of $Y$. Here, $X$ and $Y$ are random variables representing the image features and the labels respectively.

The pointwise Bayes optimal mapping is defined as [17]:

$$\phi(\mathbf{x}) \;=\; \arg\min_{\mathbf{q} \in M} R(\mathbf{x}, \mathbf{q}) \;=\; \arg\min_{\mathbf{q} \in M} \int_M s(\mathbf{y}, \mathbf{q})d\mu_{\mathbf{x}}(\mathbf{y}) + \mathcal{R}_{\text{clip}} \tag{6}$$

where $d\mu_{\mathbf{x}}$ is the conditional probability of $Y$ conditioned on $X = \mathbf{x}$ and $M$ is the output space. Following [17], we leverage kernel estimators to rewrite the adaptation problem as:

$$\phi(\mathbf{x}) \;=\; \arg\min_{\mathbf{q}} \frac{1}{NK} \sum_{i=1}^{NK} k_\beta(d(\mathbf{x}, \mathbf{S}_i))s(\mathbf{q}, \mathbf{L}_i) + \mathcal{R}_{\text{clip}} \tag{7}$$

This formulation offers a new perspective on the adaptation, where for each test point the cost function is minimized over the output space, with a weighting from each training sample. The regularization term guarantees that the obtained predictions are not far from zero-shot CLIP.

Interestingly, this formulation paves the way to considering different choices of cost functions $s$, regularization terms $\mathcal{R}_{\text{clip}}$ and kernels $k_\beta$. The consistency of the obtained estimator is discussed in [17] where the solution of 7 is obtained using a gradient descent algorithm. While this optimisation can be time consuming, there exists some cases where a closed form solution can be derived. For instance, when $s$ is the squared Euclidean distance and $\mathcal{R}_{\text{clip}} = \lambda\|\mathbf{q} - f_{\text{clip}}(\mathbf{x})\|_2^2$, we can derive the following solution:

$$\phi(\mathbf{x}) \;=\; \frac{\lambda NK}{\lambda NK + \mathcal{Z}(\mathbf{x})} f_{\text{clip}}(\mathbf{x}) + \frac{1}{\lambda NK + \mathcal{Z}(\mathbf{x})} \sum_{i=1}^{NK} k_\beta(d(\mathbf{x}, \mathbf{S}_i))\mathbf{L}_i \tag{8}$$

$$\text{where} \quad \mathcal{Z}(\mathbf{x}) = \sum_{i=1}^{NK} k_\beta(d(\mathbf{x}, \mathbf{S}_i))$$

where $\lambda$ is a regularization term that balances between the predictions of zero-shot CLIP and the local fit. The obtained closed form of the estimator in equation 7 is equivalent to Tip-Adapter up to a

scaling factor which depends on each input $\mathbf{x}$. The main difference between the two formulations in equation 2 and 8 is that the second term in equation 8 is agnostic to the training size and is query dependent.

Although the Nadaraya-Waston estimator is a nonparametric model that allows to capture any type of distribution, it is well known to suffer from poor bias at the boundaries of the training samples [36]. In the following, we propose two methods to alleviate this bias.

### 3.3 LOCAL LINEAR REGRESSION

A standard approach to alleviate the boundary bias of the NW estimator is by moving from a local constant fit to a local linear fit around each test point. Instead of using the estimate from equation 7, local linear regression (LLR) forms the local estimate $\phi(\mathbf{x}) = \tilde{\mathbf{x}}\mathbf{A}$ where $\tilde{\mathbf{x}} = \begin{bmatrix} 1 & \mathbf{x} \end{bmatrix}$ and $\mathbf{A} \in \mathbb{R}^{(d+1)c}$ minimizes the following problem:

$$\min_{\mathbf{A}} \frac{1}{NK} \sum_{i=1}^{NK} k_\beta(d(\mathbf{x}, \mathbf{S}_i)) s(\tilde{\mathbf{S}}_i \mathbf{A}, \mathbf{L}_i) + \mathcal{R}_{\text{clip}} \tag{9}$$

which is a weighted ordinary least square problem around each test point weighted by the kernel values $k_\beta(d(\mathbf{x}, \mathbf{S}_i))$. Using the same cost function as for equation 8 and using the regularization term $\mathcal{R}_{\text{clip}} = \lambda \|\tilde{\mathbf{x}}\mathbf{A} - f_{\text{clip}}(\mathbf{x})\|_2^2$, we derive a closed form solution for equation 23 as follows:

$$\phi(\mathbf{x}) = \tilde{\mathbf{x}} \left( \tilde{\mathbf{S}}^\top \mathbf{\Omega} \tilde{\mathbf{S}} + \lambda NK \tilde{\mathbf{x}}^\top \tilde{\mathbf{x}} \right)^{-1} \left( \tilde{\mathbf{S}}^\top \mathbf{\Omega} \mathbf{L} + \lambda NK \tilde{\mathbf{x}}^\top f_{\text{clip}}(\mathbf{x}) \right) \tag{10}$$

where $\mathbf{\Omega}$ is the $NK \times NK$ matrix with $i$th diagonal element as $k_\beta(d(\mathbf{x}, \mathbf{S}_i))$.

This method, that we dub henceforth LLR, boils down to fitting a local linear regression while enforcing the obtained fit to be close to the text predictions locally around the query input sample. While LLR usually improves performance, it is time consuming as one needs to fit a linear regression for each input sample. Furthermore, the parameters of LLR are solely estimated locally, which is prone to overfitting in high-dimensional problems [36]. Alternatively, one possible way to eliminate the bias of NW estimator due to the bandwidth selection without significantly increasing the time complexity is by equipping the kernel with a better distance function $d$ [36].

### 3.4 LOCAL METHODS WITH A GLOBAL METRIC

While there exists multiple strategies to construct a good metric [56; 36], using the Mahalanobis distance with the covariance matrix estimated from the training data is a simple yet well-performing one [3]:

$$d(\mathbf{x}, \mathbf{S}_i) = \|\mathbf{x} - \mathbf{S}_i\|_{\hat{\mathbf{\Lambda}}} \tag{11}$$

where $\hat{\mathbf{\Lambda}}$ is the estimated precision matrix from the few-shot samples. This metric effectively incorporates global information and captures the geometry of the space which allows to construct a better kernel function tailored to the downstream task. The classical RBF kernel corresponds to using an isotropic covariance matrix. While still being a local method, adapting the metric to the downstream task allows to incorporate global information about the underlying distribution and improves over Tip-Adapter as shown in Figure 2.

### 3.5 PROXIMAL KERNEL RIDGE REGRESSION

As can be seen in Figure 2, incorporating global information about the task through a global metric effectively outperforms both Tip-Adapter and LLR. However, the choice of the global metric for NW estimator beyond the Mahalanobis distance remains challenging [56], especially in the few-shot setting. Furthermore, despite being equipped with a global metric, the NW estimator remains in essence a local method and still lacks a global regularization. Whilst the use of a global regularization has been recently addressed in training-based methods [50], using a truly global regularization in a training-free manner remains challenging and unexplored.

These limitations highlight the necessity for regularization in this adaptation process. The main idea is to balance the need to maintain the expressive capacity of the learned functions while ensuring stability and robustness. To this end we devise two important design choices. Firstly, we restrain the hypothesis space of the learned function to be a reproducing kernel Hilbert space (RKHS). Secondly, using the RKHS norm, we introduce a proximal regularization term to ensure that the obtained solution is close to the base predictor i.e. $f_{\text{clip}}$. Thanks to the properties of the RKHS, minimizing the difference between two functions using the RKHS norm ensures that they are close pointwise. Consequently, this proximal term serves as global regularization that preserves prior knowledge from the zero-shot predictor, resulting in more robust solutions that are less prone to overfitting on the few-shot data.

Given the multi-output nature of the problem, the employed reproducing kernel $\mathbf{K}_\beta$ is a reproducing kernel for vector valued functions. The main difference is that the kernel is matrix valued. Several instances of multi-output kernels have been proposed in the literature [5; 1], with separable kernels being among the most widely used for learning vector-valued functions due to their simplicity and computational efficiency. These kernels are formulated as a product of a kernel function for the input space alone and a matrix that encodes the interactions among the outputs. Let us consider $\mathbf{K}_\beta : \mathbb{R}^D \times \mathbb{R}^D \mapsto \mathbb{R}^{N \times N}$ as a separable kernel of the form:

$$(\mathbf{K}_\beta(\mathbf{x}, \mathbf{x}'))_{j,j'} = k_\beta(\mathbf{x}, \mathbf{x}')\mathbf{B} \tag{12}$$

where $\mathbf{B}$ is a $N \times N$ symmetric and positive semi-definite matrix which captures the correlations between the outputs. A simple, yet effective choice for $\mathbf{B}$ is the identity matrix where all outputs are treated as being unrelated.

Our goal is to learn a multi-output predictor $\phi$ using the following objective:

$$\min_{\phi \in \mathcal{H}} \sum_{i=1}^{NK} \|\phi(\mathbf{S_i}) - \mathbf{L}_i\|_2^2 + \lambda \|\phi - f_{\text{clip}}\|_{\mathcal{H}}^2 \tag{13}$$

By the representer theorem [32; 22], the unique minimizer of problem 13 emerges naturally as the solution of a Kernel Ridge Regression (KRR) problem:

$$\phi = f_{\text{clip}} + \sum_{i=1}^{NK} k_\beta(\mathbf{S_i}, .)\boldsymbol{\gamma}_i \tag{14}$$

$$\text{where} \quad \boldsymbol{\gamma} = (\mathbf{I} + \frac{1}{\lambda}k_\beta(\mathbf{S}, \mathbf{S}))^{-1}(\mathbf{L} - f_{\text{clip}}(\mathbf{S}))$$

Here, $(k_\beta(\mathbf{S}, \mathbf{S}))_{i,j} = k_\beta(\mathbf{S}_i, \mathbf{S}_j)$ and $\boldsymbol{\gamma}_i \in \mathbb{R}^N$. This approach allows to map data to an infinite dimensional space. Furthermore, the regularization term allows the use of a richer model that captures the complex structure of the data while preserving its smoothness, avoiding overfitting on the few-shot data.

### 3.6 MERCER DECOMPOSITION OF KERNEL METHODS

One additional benefit of this perspective on caching methods is memory reduction which allows to overcome the necessity of storing training data. For positive definite kernel, we can leverage Mercer theorem [13] to write the kernel function as follow:

$$k_\beta(\mathbf{x}, \mathbf{x}') = \psi_\beta(\mathbf{x})\psi_\beta(\mathbf{x}^\top) \tag{15}$$

This allows us to write :

$$\sum_{i=1}^{NK} k_\beta(\mathbf{S_i}, \mathbf{x})\gamma_i = \sum_{i=1}^{NK} \psi_\beta(\mathbf{x})\psi_\beta(\mathbf{S_i})\gamma_i = \psi_\beta(\mathbf{x})[\mathbf{\Psi}_\beta^1, \dots, \mathbf{\Psi}_\beta^N] \tag{16}$$

Hence we compute prototypes per class without the need to store additional samples. However, the feature map $\psi$ may not be available in closed form and may be infinite dimensional. For shift invariant kernels like the Gaussian kernel, we leverage the Bochner's theorem following [42] to write:

$$k_\beta(\mathbf{x}, \mathbf{x}') = k(\mathbf{x} - \mathbf{x}') = \mathbb{E}_{\mathbf{w} \sim p(\mathbf{w})}(\psi_\mathbf{w}(\mathbf{x})\psi_\mathbf{w}(\mathbf{x}')^\top) \quad \text{where} \quad \psi(\mathbf{x}) = \exp(i\mathbf{x}\mathbf{w}) \tag{17}$$

where $p(\mathbf{w})$ is the Fourier transform of the kernel $k$. This formulation allows to approximate the RBF kernel with Random Fourier features (RFF). In practice, to lower the variance of the kernel the estimate and thus keep a good balance between performance and the number of RFFs we use Orthogonal Fourier Features [59].

# 4 EXPERIMENTS

In this section, we evaluate our method on multiple image classification benchmarks. We compare our results to existing training-free methods, notably Tip-Adapter [62], which is the baseline of our work. For a fair comparison, we use the same text inputs for all reported methods. We also compare to other state-of-the-art conventional methods, such as APE [67], Tip-X [53], GDA [55], CLIP [41] and CALIP [15] which proposes a parameter-free attention mechanism to improve CLIP in a zero-shot manner. We run APE with the same text templates as Tip-Adapter using their official implementation. Finally, we report the average running times on ImageNet [9] for each method using an NVIDIA RTX A6000 GPU. For completeness, we additionally provide a comparison of our method with existing state-of-the-art training-based methods.

## 4.1 DATASETS AND EVALUATION PROTOCOL

For comprehensive evaluation, we adopt 11 image classification benchmarks: ImageNet [9], Caltech101 [11], DTD [8], EuroSAT [18], FGVCAircraft [31], Flowers102 [35], Food101 [4], Oxford-Pets [40], StanfordCars [24], SUN397 [57], and UCF101 [51]. For testing the generalization ability of our method, we further test on ImageNet-Sketch [48] and ImageNet-V2 [44].

For a fair comparison with previous works, we use ResNet-50 for the visual encoder of CLIP unless mentioned otherwise. We follow two settings for our experiments. The first setting, initially introduced by [50] for training-based methods, consists of selecting the best hyperparameters of each method on ImageNet and transfer them to the other datasets and report the average performance. This setting reflects real-life scenarios where a validation set may not be available especially in a few-shot regime. The second setting follows CoOp's benchmark [66] where validation shots are used to select the hyperparameters and evaluate the results on the full test set.

## 4.2 EXPERIMENT RESULTS AND ANALYSIS

### 4.2.1 COMPARISON WITH ALTERNATIVES TO TIP-ADAPTER

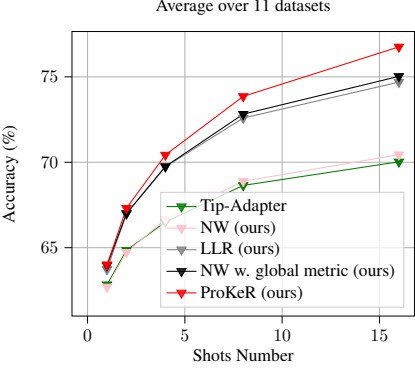

Figure 2: **Average performance for different methods on 11 image classification datasets.**

We compare in Figure 2 our result when improving Tip-Adapter using kernel based approaches across 11 datasets. Our reformulation in equation 8 outperforms Tip-Adapter for 8 and 16 shots and maintains the same level of performance in the lower shot setting. We argue that this is due to the fact that our reformulation is agnostic to the training size. Debiasing the NW estimator using our regularized LLR in equation 23 significantly improves the performance as a first order polynomial fit compared to the constant fit of the NW estimator. Additionally, the use of the Mahalanobis distance

as global metric for NW estimator further increases the performance especially with more shots as the estimation of the precision matrix from the training set becomes more accurate. Furthermore, by introducing a proximal regularization in the RKHS, our method ProKeR significantly outperforms Tip-Adapter as well as all the proposed alternatives especially with more shots.

### 4.2.2 COMPARISON WITH STATE-OF-THE-ART

We report in Table 1 the performance of different methods in a realistic and practical validation-free experimental setting. First, our method ProKeR outperforms on average existing alternatives by a far margin (2.04% compared to the second best), especially in the low shot regimes (1, 2 and 4) and is only outperformed in the 8 shot setting. Furthermore, when using a Polynomial kernel (defined in Table 4), our method sets a new standard in the 16 shots regime.

Table 1: Average performance on 11 classification datasets for different shots. Hyperparameters are transfered from ImageNet.

| Shots | 1 | 2 | 4 | 8 | 16 | Average |
|---|---|---|---|---|---|---|
| Tip-Adapter [62] | 58.86 | 60.33 | 61.49 | 63.15 | 64.61 | 61.68 |
| APE [67] | 60.09 | 62.33 | 65.36 | 67.95 | 69.89 | 65.12 |
| GDA [55] | 57.49 | 63.43 | 66.68 | **72.46** | 75.12 | 67.03 |
| ProKeR (Polynomial) (ours) | 62.39 | 65.59 | 68.05 | 71.81 | **75.82** | 68.72 |
| ProKeR (RBF) (ours) | **63.13** | **66.31** | **68.64** | 72.15 | 75.12 | **69.07** |

In the CoOp's benchmark where validation shots are used to tailor hyperparamters for each dataset, our method ProKer surpasses existing training-free methods as shown in Table 2. These superior results fully validate the significance of using the global regularization in the RKHS.

Table 2: Performance on 11 different classification datasets (CoOp's benchmark).

| Shots | 1 | 2 | 4 | 8 | 16 | Average |
|---|---|---|---|---|---|---|
| GDA [55] | 62.19 | 66.19 | 69.77 | 73.30 | 76.04 | 69.49 |
| Tip-Adapter [62] | 62.83 | 64.84 | 66.50 | 68.64 | 70.02 | 66.56 |
| Tip-Adapter [62] with RFF | 62.77 | 64.58 | 66.29 | 68.15 | 69.60 | 66.27 |
| APE [67] | **64.43** | 66.48 | 68.66 | 70.75 | 72.81 | 68.62 |
| APE [67] with RFF | 64.16 | 66.11 | 68.76 | 70.37 | 72.02 | 68.28 |
| ProKeR (ours) | 64.01 | **67.31** | **70.42** | **73.85** | **76.75** | **70.46** |
| ProKeR with RFF (ours) | 64.01 | 67.12 | 70.28 | 73.61 | 76.44 | 70.29 |

### 4.2.3 GENERALIZATION ABILITY

Table 3: **Robustness to distribution shift of different methods for 16 shots.**

| Datasets | Source | Target | | | | |
|---|---|---|---|---|---|---|
| | ImageNet [9] | -V2 [44] | -Sketch [48] | -A [20] | -R [19] | Average |
| Zero-Shot CLIP [41] | 60.33 | 53.27 | 35.44 | 23.61 | 60.42 | 46.61 |
| Tip-Adapter [62] | 61.43 | 54.13 | 35.71 | **23.63** | 60.41 | 47.06 |
| APE [67] | 62.60 | 54.93 | 35.41 | 22.95 | 59.90 | 47.15 |
| GDA [55] | 63.82 | 55.35 | 34.32 | 19.53 | 55.56 | 45.71 |
| ProKeR (Polynomial) (Ours) | **64.66** | **56.11** | **36.08** | 23.27 | 60.55 | **48.13** |
| ProKeR (Ours) | 64.45 | 56.02 | **36.08** | 23.37 | **60.59** | 48.10 |

In Table 3, we test the generalization ability of the different methods on out-of-distribution datasets. Notably, the shots are drawn from ImageNet and the test set is drawn from either ImageNet-V2 or

ImageNet-Sketch. ProKeR achieves state-of-the-art performance for training-free methods on both in-distribution and out-of-distribution datasets.

### 4.3 KERNEL ABLTATION

So far, we have performed our analysis using the RBF kernel, a commonly used kernel in the kernel literature and in cache-based methods. Nevertheless, through the lens of our kernel perspective on cache-based, different kernels can be considered ranging from a linear kernels to more elaborate ones. We perform in Table 4 an ablation study where we discuss different kernel choices. Besides the RBF kernel, we consider three commonly used kernels: the Linear kernel, the Epanechnikov kernel and the Polynomial kernel. The RBF kernel outperforms the linear, Epanechnikov and the polynomial kernels. Using the RBF kernel allows us to project data into an infinite dimensional space which captures more complex relationships.

| Kernel | $k(\mathbf{x}, \mathbf{x}')$ | Accuracy |
|---|---|---|
| Linear | $\mathbf{x}\mathbf{x}'^{\top}$ | 72.34 |
| Epanechnikov | $\frac{3}{4}\left(1 - \|\mathbf{x} - \mathbf{y}'\|_2^2\right)$ | 74.43 |
| Polynomial | $\left(\mathbf{x}\mathbf{y}'^{\top}\right)^2$ | 76.61 |
| RBF | $\exp(-\frac{\beta}{2}\|(\mathbf{x} - \mathbf{y}'\|_2^2))$ | **76.75** |

Table 4: Kernel Ablation for 16 shots on 11 datasets on CoOp's benchmark.

### 4.4 ABLATION ON CLIP ARCHITECTURES

In Table 5, we report the performance of training-free methods using different backbones on ImageNet for 16 shots. Our method consistently performs better than the alternatives across all architectures. While all methods improve with a more capable architectures, the gap between our method and the second best one (APE) remains stable.

Table 5: **Average performance on 16-shot ImageNet with different backbones.**

| Models | ResNet-50 | ResNet-101 | ViT-B/32 | ViT-B/16 |
|---|---|---|---|---|
| Zero-shot CLIP [41] | 60.33 | 62.53 | 63.80 | 68.73 |
| Tip-Adapter [62] | 61.43 | 64.08 | 65.18 | 70.25 |
| APE [67] | 62.60 | 65.61 | 66.31 | 71.37 |
| ProKeR (Ours) | **64.45** | **67.39** | **68.12** | **73.25** |

### 4.5 ADRESSING MEMORY LIMITATIONS OF CACHE-BASED METHODS

In Table 2, we report the performance when using Random Fourier features to alleviate the memory limitations of cache-based methods. Using RFFs, we're able to drastically reduce the memory print of caching methods while maintaining almost the same performance across different shots. Our method when combined with RFFs still maintains state-of-the-art performance.

### 4.6 RUNNING TIMES AND MEMORY REQUIREMENTS

Next, we report running time (train and test) for different methods as well as the memory requirements for each method. Our method is on part with Tip-Adapter and GDA in term of speed. Note that APE runs a feature selection step which takes additional time to run, the third position. On the other hand, training-based methods are orders of magnitude slower. Regarding the memory complexity, ProKeR stores the training shots similarly to APE and Tip-Adapter. However, when using Random Fourier features, our method does need to store additional training samples.

Table 6: **Running times on ImageNet for 16 shots.** All experiments are performed on a RTX A6000 GPU. For each method we report the memory requirements. $D_1$ is the inner dimension of the MLP used in Clip-Adapter [12] and $T$ is the number of text tokens in CoOp [66]. $R$ is the number of Fourier features used tp approximate the RBF kernel.

| Methods | Overall Time | Train | Memory requirements |
|---------|-------------|-------|---------------------|
| CoOp[66] | $\sim 17h$ | ✓ | $N \times T \times D$ |
| Clip-Adapter[12] | $\sim 40min$ | ✓ | $(2N + D_1) \times D$ |
| CrossModal-LP[28] | $\sim 3min$ | ✓ | $N \times D$ |
| Standard LP[41] | $\sim 3min$ | ✓ | $N \times D$ |
| Tip-adapter-F[62] | $\sim 7min$ | ✓ | $N \times K + N \times (K+1) \times D$ |
| Tip-Adapter[62] | 2.1s | ✗ | $N \times K + N \times (K+1) \times D$ |
| APE [67] | 24.6s | ✗ | $N \times K + N \times (K+1) \times D$ |
| GDA [55] | 1.6 s | ✗ | $2 \times N \times D$ |
| ProKeR (Ours) | 4.7s | ✗ | $N \times K + N \times (K+1) \times D$ |
| ProKeR with RFF (Ours) | 4.2s | ✗ | $N \times (D+R)$ |

## 5 LIMITATIONS & FUTURE WORK

Based on our analysis, caching methods can be understood as local nonparametric regressors. These methods lack a global regularization from zero-shot CLIP which limits their generalization ability when performing adaptation. On the other hand, global methods may lack the flexibility when dealing with complex data. In the future work, we will explore how both local and global methods can be combined to benefit from the best of both worlds. Furthermore, our formulation of caching methods as a Nadaraya-Watson estimator offers multiple options for the choice the regularization term, the metric used in the kernel as well as the bandwidth selection [39]. These choices constitute different ways to reduce the bias-variance trade-off inherent to these methods [16].

## 6 CONCLUSION

In this paper, we propose a theoretical understanding of Tip-Adapter, a training-free caching-based method. Our analysis suggests that Tip-Adapter is a local nonparametric regression that has well-known bias limitation. We propose multiple angles of improvement that has shown significant amelioration over Tip-Adapter's baseline. Subsequently, we demonstrate that incorporating global information in a training-free method can be achieved using a global regularization in a reproducing kernel Hilbert space (RKHS), which conclusively further improves the state-of-the-art for training-free methods.

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

# A  SUPPLEMENTAL MATERIAL

## A.1  DETAILED DERIVATIONS

### A.1.1  NADARAYA-WATSON ESTIMATOR

We first derive the solution of the adaptation problem for the Nadaraya-Waston estimator in equation 8. The adaptation problem writes as:

$$\phi(\mathbf{x}) = \arg\min_{\mathbf{q}} \frac{1}{NK} \sum_{i=1}^{NK} k_\beta(d(\mathbf{x}, \mathbf{S}_i)) \|\mathbf{q} - \mathbf{L}_i\|_2^2 + \|\mathbf{q} - f_{\text{clip}}(\mathbf{x})\|_2^2 \tag{18}$$

The derivation of the solution of equation 18 is as follows:

$$\text{Let} \quad \mathcal{L} = \frac{1}{NK} \sum_{i=1}^{NK} k_\beta(d(\mathbf{x}, \mathbf{S}_i)) \|\mathbf{q} - \mathbf{L}_i\|_2^2 + \|\mathbf{q} - f_{\text{clip}}(\mathbf{x})\|_2^2 \tag{19}$$

$$\frac{\partial \mathcal{L}}{\partial \mathbf{q}} = 0 \Rightarrow \frac{1}{NK} \sum_{i=1}^{NK} k_\beta(d(\mathbf{x}, \mathbf{S}_i)) \left(\mathbf{q} - \mathbf{L}_i\right) + \lambda\mathbf{q} - \lambda f_{\text{clip}}(\mathbf{x}) = 0 \tag{20}$$

$$\Rightarrow \mathbf{q} \left(\lambda NK + \sum_{i=1}^{NK} k_\beta(d(\mathbf{x}, \mathbf{S}_i))\right) = \lambda NK f_{\text{clip}}(\mathbf{x}) + \sum_{i=1}^{NK} k_\beta(d(\mathbf{x}, \mathbf{S}_i))\mathbf{L}_i \tag{21}$$

$$\Rightarrow \mathbf{q} = \frac{\lambda NK}{\lambda NK + \sum\limits_{i=1}^{NK} k_\beta(d(\mathbf{x}, \mathbf{S}_i))} f_{\text{clip}}(\mathbf{x}) + \frac{1}{\lambda NK + \sum\limits_{i=1}^{NK} k_\beta(d(\mathbf{x}, \mathbf{S}_i))} \sum_{i=1}^{NK} k_\beta(d(\mathbf{x}, \mathbf{S}_i))\mathbf{L}_i \tag{22}$$

### A.1.2  LOCAL LINEAR REGRESSION

Here, we detail the derivation of the solution of the local linear regression (LLR) in equation 10. Let $\tilde{\mathbf{x}} = [1 \quad \mathbf{x}]$ and $\mathbf{A} \in \mathbb{R}^{(d+1)c}$ which minimizes the following problem:

$$\min_{\mathbf{A}} \frac{1}{NK} \sum_{i=1}^{NK} k_\beta(d(\mathbf{x}, \mathbf{S}_i)) \|\tilde{\mathbf{S}}_i \mathbf{A} - \mathbf{L}_i\|_2^2 + \lambda\|\tilde{\mathbf{x}}\mathbf{A} - f_{\text{clip}}(\mathbf{x})\|_2^2 \tag{23}$$

Let $\mathbf{\Omega}$ be the $NK \times NK$ matrix with $i$th diagonal element as $k_\beta(d(\mathbf{x}, \mathbf{S}_i))$. The derivation is as follows:

$$\text{Let} \quad \mathcal{L} = \frac{1}{NK} \mathbf{\Omega} \|\tilde{\mathbf{S}}\mathbf{A} - \mathbf{L}\|_2^2 + \lambda\|\tilde{\mathbf{x}}\mathbf{A} - f_{\text{clip}}(\mathbf{x})\|_2^2 \tag{24}$$

$$\frac{\partial \mathcal{L}}{\partial \mathbf{A}} = 0 \Rightarrow \frac{1}{NK} \tilde{\mathbf{S}}^\top \mathbf{\Omega} \left(\tilde{\mathbf{S}}\mathbf{A} - \mathbf{L}\right) + \lambda\tilde{\mathbf{x}}^\top \left(\tilde{\mathbf{x}}\mathbf{A} - f_{\text{clip}}(\mathbf{x})\right) = 0 \tag{25}$$

$$\Rightarrow \left(\tilde{\mathbf{S}}\mathbf{\Omega}\tilde{\mathbf{S}} + \lambda NK\tilde{\mathbf{x}}^\top\tilde{\mathbf{x}}\right) \mathbf{A} = \tilde{\mathbf{S}}^\top\mathbf{\Omega}\mathbf{L} + \lambda NK\tilde{\mathbf{x}}^\top f_{\text{clip}}(\mathbf{x}) \tag{26}$$

$$\Rightarrow \mathbf{A} = \left(\tilde{\mathbf{S}}\mathbf{\Omega}\tilde{\mathbf{S}} + \lambda NK\tilde{\mathbf{x}}^\top\tilde{\mathbf{x}}\right)^{-1} \left(\tilde{\mathbf{S}}^\top\mathbf{\Omega}\mathbf{L} + \lambda NK\tilde{\mathbf{x}}^\top f_{\text{clip}}(\mathbf{x})\right) \tag{27}$$

## A.2 SENSITIVITY ANALYSIS OF $\lambda$

We analyze the sensitivity of $\lambda$ in Table 7. We compute the average value for each dataset and study the effect of varying its value. Overall, the value of lambda is quite stable in a range of 1/3 of its value up to 3 times its value with only a drop of 1.2% accuracy. Varying lambda up to a fifth or 5 times its value only leads to a drop of 3%.

Table 7: Sensitivity Analysis of $\lambda$ on 11 datasets for 16-shots.

| | $\lambda \times 5$ | $\lambda \times 4$ | $\lambda \times 3$ | $\lambda \times 2$ | $\lambda$ | $\lambda \times 2$ | $\lambda \times 3$ | $\lambda \times 4$ | $\lambda \times 5$ | ProKeR |
|---|---|---|---|---|---|---|---|---|---|---|
| Average | 73.17 | 74.23 | 75.24 | 76.27 | 76.58 | 75.85 | 75.11 | 74.45 | 73.84 | 76.75 |

## A.3 COMPARISON PER DATASET

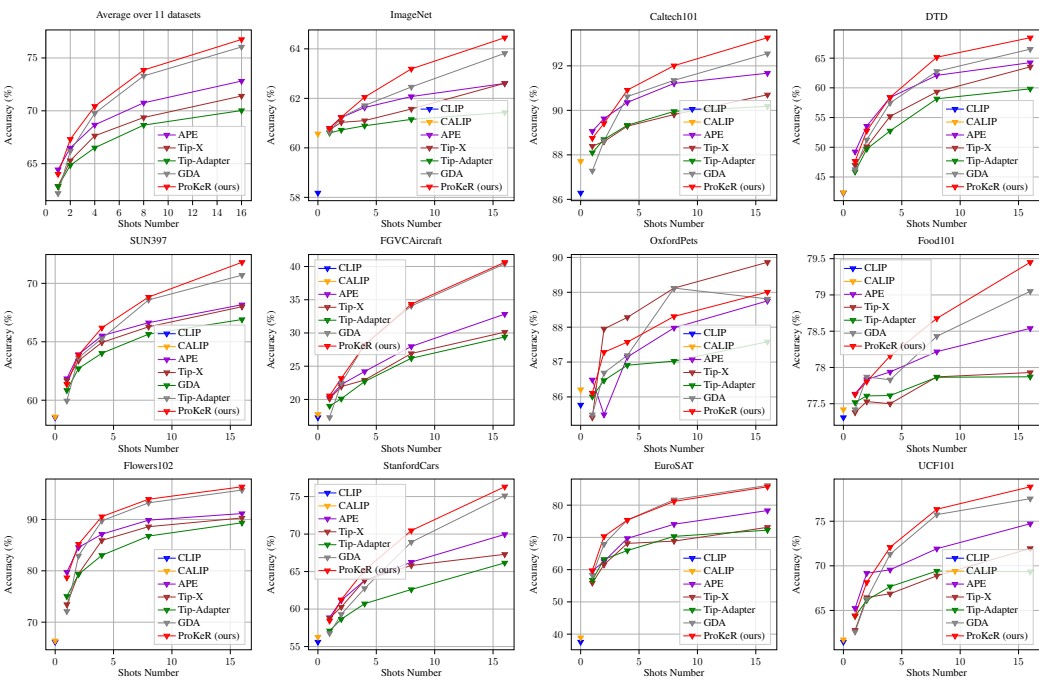

Figure 3: **Few-shot Performance of Training-free Methods** on 11 image classification datasets (CoOp's benchmark).

## A.4 COMPARISON WITH TRAINING-BASED METHODS

Table 8: Performance on 11 classification datasets for 16 shots. Hyperparameters are transfered from ImageNet.

| Method | ImageNet | Caltech101 | OxfordPets | StanfordCars | Flowers102 | Food101 | FGVC | SUN397 | DTD | EuroSAT | UCF101 | Average |
|---|---|---|---|---|---|---|---|---|---|---|---|---|
| Tip-Adapter-F [62] | 62.27 | 91.22 | 85.43 | 69.56 | 91.18 | 74.65 | 29.32 | 68.90 | 64.56 | 76.55 | 71.81 | 71.40 |
| CrossModalLP [28] | 52.90 | 92.77 | 87.48 | 75.44 | 95.20 | 77.14 | 33.30 | 70.56 | 66.92 | 82.03 | 76.40 | 73.65 |
| TaskRes [60] | 60.85 | 93.09 | 86.28 | 75.38 | 96.14 | 75.43 | 36.53 | 68.43 | 65.88 | 83.70 | 76.96 | 74.42 |
| APE-T [67] | 63.06 | 91.83 | 87.93 | 70.32 | 93.93 | 77.65 | 34.17 | 64.47 | 64.77 | 81.94 | 76.11 | 68.28 |
| CLAP [50] | **65.02** | 91.93 | **88.51** | 75.12 | **94.21** | 78.55 | 33.59 | 70.78 | 66.41 | 80.07 | 76.29 | 74.57 |
| ProKeR (Polynomial) (ours) | 64.66 | **93.36** | 88.14 | **75.68** | 92.84 | **79.26** | **37.37** | **71.47** | 67.43 | **85.37** | **78.50** | **75.82** |
| ProKeR (ours) | 63.77 | 93.23 | 88.15 | 74.58 | 90.62 | 79.14 | 35.33 | 71.44 | **67.49** | 84.49 | 78.00 | 75.11 |

We report in Table 8 the comparison of our method with training-based methods. While being training-free, our method ProKeR outperforms both existing training-based methods on 8 out of 11 datasets and outperforms the second best on average by 1.25%. This emphasizes the effectiveness of incorporating a global regularization using the zero-shot predictor in a reproducing kernel Hilbert space (RKHS).

