# OpenReview forum: "A Kernel Perspective on Training-Free Few-Shot Adaptation of Large Vision-Language Models"
_ICLR.cc/2025/Conference — ICLR 2025 Conference Withdrawn Submission_

### Official Review · Reviewer_o6Ez · 2024-10-26

**Soundness:** 3
**Presentation:** 3
**Contribution:** 3
**Rating:** 5
**Confidence:** 4

**Summary:**

This paper introduces a kernel-based method to understand and enhance the training-free adaptation methods (Tip-adapter).  The authors provide a novel analysis of the Tip-Adapter method from a kernel perspective and identify that Tip-Adapter functions as a modified Nadaraya-Watson estimator. Then, the authors propose ProKeR and RKHS proximal regularizers. Extensive experiments are conducted and  the results validate the effectiveness of the proposed ideas.

**Strengths:**

- The overall presentation is clear and easy to follow.

- This paper analyses Tip-Adapter from the kernel-based perspective and provides a novel theoretical foundation, which provides some insights for understanding training-free adapting methods.

- Based on the understanding, the authors further propose ProKeR, which leverages the RKHS framework to introduce global regularization, addressing biases present in local nonparametric methods like Tip-Adapter.

- ProKeR achieves the state-of-the-art for training-free few-shot adaptation under two settings.

**Weaknesses:**

- Reproducibility: Authors should report the values of the hyperparameters in the implementation part. Moreover, in CoOp's setting, the ImageNet dataset has no validation dataset, how do authors select hyper-parameter?

- What is validation shots? It is validation set or shots from validation set? If it is validation set, it is better to name it as validation set to avoid confusion.

- The implementation part missed many details. For example, did authors follow CoOp to run 3 times for each experiment and report the average results? How did authors use data augmentation? Since authors did not provide code, they should clarify these clearly for reproducibility.

- Table 5 missed the results of GAD.

- I understand that this paper is mainly for training-free settings. However, I am concerned about the practical usage. If we are provided with K-shot samples, why do we only use training-free methods without fine-tuning, as fine-tuning usually leads to much better results (as shown in Tip-Adapter)? Could authors further discuss how to develop ProKeR with fine-tuning (like Tip-Adapter-F) or could we combine ProKeR with fine-tuning methods to further boost their performances?

- ProKeR seems sensitive to different kernels. Could authors further discuss why specific kernels have better results for better understanding?

- An important goal of fine-tuning is to maintain the generalization ability of CLIP. Could authors report the base-to-new setting results as done in GDA to demonstrate the generalization ability of ProKeR?

- It seems that ProKeR only surpasses GDA by a little margin on CoOp's datasets, but with nearly 3 times of inference time. Could authors further discuss the advantages of ProKeR over GDA?

- Hyperparameter $\lambda$ is an important factor of the proposed method, but the sensitivity analysis is placed in the appendix. It is suggested to place this part in the main paper part. Moreover, it is strange that authors use $\lambda$ instead of true values for sensitivity analysis, could authors specify the values of $\lambda$ for analysis for better understanding?

**Questions:**

See weakness.
I am happy to revise my score if authors could address my concerns.

---

### Official Review · Reviewer_hUi6 · 2024-10-31

**Soundness:** 4
**Presentation:** 3
**Contribution:** 3
**Rating:** 5
**Confidence:** 3

**Summary:**

This paper explores training-free methods like Tip-Adapter from a kernel perspective, providing theoretical analysis and algorithmic innovations. By categorizing caching methods as Local adapters, the paper proposes a new global adapter that learns a proximal regularizer in a reproducing kernel Hilbert space. Experiments are conducted on standard benchmarks.

**Strengths:**

1. The paper provides a theoretical explanation from a kernel perspective for methods like Tip-Adapter, validating the effectiveness of existing approaches.
2. Based on this theoretical framework, a new algorithmic design that incorporates global information is proposed.
3. There are improved results on standard benchmarks reported, illustrating the practical efficacy of the proposed method.

**Weaknesses:**

1. The citation format in the paper does not align with the ICLR template; the paper uses numerical references, whereas ICLR prefers author-name citations (e.g., CLIP by Radford et al., 2021).
2. The relationship between equations (2) and (3) is unclear. Equation (3) includes a denominator that is absent in equation (2); how should these be corresponded and understood?
3. The concept of "global" as opposed to "local" is confusing in this context. Familiar Tip-Adapter methods use weighted few-shot samples to predict test sample outcomes, utilizing all training samples, which seems global in nature. What explicit meanings are assigned to "global" and "local" in this paper?
4. The focus of the paper on training-free CLIP adaptation could benefit from a discussion of recent related works such as [1].

[1]Dual memory networks: A versatile adaptation approach for vision-language models, CVPR2024

**Questions:**

see weaknesses

---

### Official Review · Reviewer_7dUy · 2024-10-31

**Soundness:** 3
**Presentation:** 2
**Contribution:** 2
**Rating:** 5
**Confidence:** 4

**Summary:**

This paper focuses on the few shot adaptation of large scale vision-language models (VLMs), particularly for CLIP-based cache model.
The paper first rethinks the Tip-Adapter from the perspective of kernel, and then proposes a training-free method called ProKeR (Proximal Kernel ridge Regression) based on such kernel perspective.
Extensive experiments reveal that ProKeR achieves a new SOTA on few-shot classification benchmarks.

**Strengths:**

1.The paper provides a novel theoretical framework for better understanding the caching models, coming with detailed derivation.

2.The proposed ProKeR achieves competitive performance on most few-shot classification datasets.

**Weaknesses:**

1.Rigorousness of theory:

①Some claims have no theoretical basis, e.g., in line 201 & 277-279, the author claims that the regularization preserves prior knowledge, and predictions are not far from the zero-shot predictor, which is better for few-shot classification.
But why predictions close to zero-shot predictor will be better, can you provide some theoretical evidence?

Relatively, there are some contrasting methods, like AMU-Tuning, which does not perform regularization or require to be close to zero-shot predictor, but it gets better results.
Moreover, after introducing features from extra model, AMU-Tuning’s prediction will be further away from zero-shot predictor, but it achieves much better performance (about 5% higher than author’s on ImageNet before fine-tuning).
The important thing is that a single model’s prediction have low performance, but when their prediction are combined, it will achieve a high performance.

Does the conclusion of AMU-Tuning indicate that CLIP's zero-shot prediction is sub-optimal? This seems contradict to author's claims and motivation.
Can author compare their methods, and explain why AMU-Tuning’s prediction is far from CLIP's zero-shot predictor, but achieves extremely better results?
Is this reveals CLIP's zero-shot prediction is sub-optimal?
Or is it because your method does not fully capture novel knowledge from support set?
Is this related to overfitting on the few-shot data?
If CLIP's zero-shot prediction is indeed sub-optimal, is it reasonable for author to believe that the prediction should be close to CLIP's zero-shot prediction?

②There are many estimates and approximations in paper, but without providing any analysis of errors. However, in extreme low-data situations, such errors may be fatal.
For example, from equation 6 to equation 7, you have quantified the continuous mapping into a discrete form. But in fact, for few-shot problem, the value of K is very small.
It is suggested to calculate the quantization errors and analyze the impact of K on such errors.


2.The contribution and practical value.
The paper integrates large number of components like RBF, LLR, local method, global metric, RKHS, RFF, Polynomial, etc., making the mechanism very complex and cumbersome, but the advantages compared to SOTAs are not obvious.
In summary, it has complex mechanism and non-competitive resource consumption, but does not achieve much better performance, i.e., on CoOp’s benchmark, only a bit higher than GDA.
A comprehensive component ablation should be conducted for readers to clearly identify the main components and their effect.


3.Weakness on writing.
The paper lists a series of components and proprietary terms, some of which are secondary, but the author only describes them one by one without highlighting the core components.
And some components, like RKHS, have no introduction or contextual support, making it difficult for readers to understand their principles and why author need to use them.
The mentioned problems make the key points of paper not clear and prevent readers to understand it.
It is more like a document introducing the process rather than explaining an interesting method.
I think the authors should reorganize the paper with a high-level overview, a clear pipeline illustration and a road-map of their method early in the paper.
And they can consider including the descriptions of the unimportant components into the appendix to highlight their core framework.



Reference: AMU-Tuning: Effective Logit Bias for CLIP-based Few-shot Learning

**Questions:**

1.I think the paper has shortcoming in theory or motivation, which needs to be researched and revised.

2.The insight is inspiring, but the author should simplify their methods and improve the writing to make the paper looks more appealing.

---

### Official Review · Reviewer_6W9D · 2024-11-04

**Soundness:** 3
**Presentation:** 3
**Contribution:** 2
**Rating:** 6
**Confidence:** 4

**Summary:**

The authors propose ProKeR, a training-free few-shot adaptation method for vision-language models like CLIP that leverages a kernel perspective. Building on Tip-Adapter, they frame the caching-based adaptation as a kernel regression problem, introducing a proximal regularization in a reproducing kernel Hilbert space to capture both local and global features. ProKeR integrates global regularization with few-shot data, incorporating global information in local adapters. Extensive experiments on both fine-grained and OOD datasets demonstrate that ProKeR consistently outperforms existing methods.

**Strengths:**

- ProKeR’s kernel-based approach is theoretically grounded and provides a fresh perspective for improving Tip-Adapter’s framework.

- ProKeR effectively enhances local adapters with global information, striking a balance between local adaptability and global regularization to prevent overfitting.

-  The method is both memory- and computation-efficient, leveraging Random Fourier Features and a closed-form solution in kernel ridge regression to reduce resource demands.

**Weaknesses:**

- ProKeR is only tested on training-free few-shot adaptation, which restricts its scope. Exploring the effects of loosening or reinforcing this constraint could enhance the method's practical relevance. For instance, would additional computational resources improve ProKeR’s performance, for example by training? Alternatively, if no few-shot samples were available, could it still operate effectively? Additionally, recent methods like DMN[1], which utilize an attention-based cache for both zero-shot and few-shot adaptation in training and training-free contexts, are absent from ProKeR’s comparison set.

- ProKeR aims to enhance few-shot adaptation by aligning few-shot and zero-shot features in RKHS. However, the paper does not adequately assess the effectiveness of this alignment. A deeper analysis of feature alignment, such as using metrics like cosine similarity between few-shot and zero-shot features or visualizations via t-SNE, could bolster the credibility of the method and provide clearer insights into how effectively ProKeR bridges this feature gap.

- ProKeR  is evaluated for few-shot adaptation only in the experiments. It is not explicitly discussed if the method could handle zero-shot adaptation instead.

[1] Zhang, Yabin, et al. "Dual memory networks: A versatile adaptation approach for vision-language models." Proceedings of the IEEE/CVF conference on computer vision and pattern recognition. 2024.

**Questions:**

- Additional comparisons with more recent few-shot adaptation methods, e.g. DMN, is necessary.

- Discussion and/or experiments on zero-shot adaptation is important to fully explore the boundary of the proposed method.

---

### Note · Authors · 2024-11-15

I have read and agree with the venue's withdrawal policy on behalf of myself and my co-authors.